# Varieties Of Evolved Forms Of Consciousness, Including Mathematical Consciousness

**DOI:** 10.3390/e22060615

**Published:** 2020-06-02

**Authors:** Aaron Sloman

**Affiliations:** School of Computer Science, University of Birmingham, Birmingham B15 2TT, UK; a.sloman@cs.bham.ac.uk

**Keywords:** AI, Immanuel Kant, meta-configured genome, altricial/precocial, biological evolution, development, internal languages, mathematical consciousness, robotic consciousness, spatial awareness, varieties of consciousness

## Abstract

I shall introduce a complex, apparently unique, cross-disciplinary approach to understanding consciousness, especially ancient forms of mathematical consciousness, based on joint work with Jackie Chappell (Birmingham Biosciences) on the Meta-Configured Genome (MCG) theory. All known forms of consciousness (apart from recent very simple AI forms) are products of biological evolution, in some cases augmented by products of social, or technological evolution. Forms of consciousness differ between organisms with different sensory mechanisms, needs and abilities; and in complex animals can vary across different stages of development before and after birth or hatching or pupation, and before or after sexual and other kinds of maturity (or senility). Those forms can differ across individuals with different natural talents and environments, some with and some without fully functional sense organs or motor control functions (in humans: hearing, sight, touch, taste, smell, proprioception and other senses), along with mechanisms supporting meta-cognitive functions such as recollection, expectation, foreboding, error correction, and so forth, and varying forms of conscious control differing partly because of physical differences, such as conjoined twins sharing body parts. Forms of consciousness can also differ across individuals in different cultures with different shared theories, and social practices (e.g., art-forms, musical traditions, religions, etc.). There are many unanswered questions about such varieties of consciousness in products of biological evolution. Most of the details are completely ignored by most philosophers and scientists who focus only on a small subset of types of human consciousness—resulting in shallow theories. Immanuel Kant was deeper than most, though his insights, especially insights into mathematical consciousness tend to be ignored by recent philosophers and scientists, for bad reasons. This paper, partly inspired by Turing’s 1952 paper on chemistry-based morphogenesis, supporting William James’ observation that all known forms of consciousness must have been products of biological evolution in combination with other influences, attempts to provide (still tentative and incomplete) foundations for a proper study of the variety of biological and non-biological forms of consciousness, including the types of mathematical consciousness identified by Kant in 1781.

## 1. Introduction

William James ‘Are we automata?’ [1] (cited by Adrian Kent at the 2019 MOC conference), is not easy to read but seems to me to correctly characterise consciousness as having evolved since the earliest life forms (which “has been slowly evolved in the animal series, and resembles in this all organs that have a use”). He had read and been influenced by Huxley’s struggles to account for consciousness [2] (which had several different editions, with revisions). Several other 19th Century scientists and philosophers discussed connections between biological evolution and various aspects of consciousness, while struggling to understand in detail how physiological mechanisms (physical/chemical parts of living organisms) and processes could produce something so different as conscious states and processes. Darwin’s work on Emotions [3] is another example, implicitly regarding consciousness as a product (or a collection of products) of biological evolution, with many different forms, functions, and mechanisms, whose variety is lost in simple slogan-definitions that have become popular recently (e.g., “what it’s like to be/feel…”), usually focused on human sensory self-consciousness, ignoring all other types, such as wanting, fearing, wondering whether, regretting, having niggling doubts, struggling to find a flaw in a computer program or mathematical proof, and many others (some included below), and also ignoring other species, such as the house-fly whose consciousness of the rapidly approaching fly-swatter enables it to switch very rapidly from feeding to successful escape behaviour. In Reference [4] Chrisley and I described additional features needing to be explained, and offered a partial explanation of “privacy” of some kinds of consciousness based on causal indexicality. But that left many other things, including mathematical consciousness, unexplained.

Types, functions and mechanisms of consciousness in this broad sense vary not only across evolutionary timescales in different species, but also change dramatically during individual development (e.g., from new-born infant, or pre-born foetus, to professor of mathematics, or brick-layer, or …). Any deep theory of consciousness should explain what makes such evolutionary and developmental transitions possible, and what the biological consequences of those transitions are.

There is no reason to assume that all those functions and mechanisms occur only in humans. In an individual developing human, there is considerable diversity of (biological, and later social) functions of consciousness and also diversity of requirements for explanatory mechanisms, including sub-personal control mechanisms shared with other species. Simpler varieties must have existed in our evolutionary ancestors and many exist now in other species. More complex varieties include co-consciousness in interacting individuals. I don’t know whether the ways in which humans of various ages and other animals (such as dogs) share playful interaction make use of ancient products of a shared evolutionary history, or depend only on recently co-evolved mechanisms.

Not all forms of human consciousness are available to all humans. I doubt that there ever has been or ever will be another human capable of all the detailed states of consciousness experienced by a great composer such as Beethoven, although the process of performing the products (e.g., a piano sonata) includes opportunities for a different kind of creativity.

I know from personal experience that there are mathematical thinkers whose experiences of understanding complex mathematical structures and forms of reasoning far exceed what I am capable of Any deep and general theory of human consciousness should be able to explain what the mechanisms involved in that diversity are and how they originated.

Human (or more generally animal) consciousness involves far more than typical computational models of consciousness that take in a sensory array of measurements (or a collection of such arrays) and produce a new data-structure with labels attached, including possibly a summary label. For example, Fearful consciousness of a charging bull can include a complex collection of reflex changes in motivation and muscular control processes with many automatic physiological processes preparing for, and then producing, intense muscular exertion in parallel with attempting to select the best escape route. That mixture is unlikely to occur during exercises of mathematical consciousness, although consciousness of missing an aspect of a mathematical problem can produce a different kind of intense, purely intellectual, exertion!

I shall summarise a subset of the implications of viewing many types of consciousness as products of biological evolution concerned with information processing functions, including control functions, and types of sub-personal consciousness that are ignored by most recent philosophers and scientists studying consciousness, who tend to focus on immediately accessible features of their own consciousness (including striking examples of illusions such as those presented in Frisby’s [5]). Recent discussions of consciousness generally ignore the huge diversity of phenomena involving consciousness (including seeing a door handle while sleep-walking, or struggling to avoid falling when tripping over an object) and ancient forms of mathematical consciousness pointed out by Immanuel Kant [6] over two centuries ago, including consciousness of a geometric fact produced by understanding a diagrammatic proof, such as Mary Pardoe’s proof of The triangle sum theorem in http://www.cs.bham.ac.uk/research/projects/cogaff/misc/triangle-sum.html.

A complete survey would fill many volumes. I can offer only a sketchy, high-level overview, including aspects of consciousness that are now often ignored, including the diversity of types of consciousness, although many were studied by pre-20th Century researchers.

A key assumption in this work is that a distinctive feature of all forms of life is use of *information*—not in the *syntactic* sense introduced by Claude Shannon [7] (a terminological decision that confused many philosophers and scientists, although Shannon wasn’t confused) but the much older sense of “information content”, sometimes labelled “semantics”, already familiar to novelist Jane Austen over a century before Shannon, as explained in http://www.cs.bham.ac.uk/research/projects/cogaff/misc/austen-info.html. Of course, that concept of information (“Austen information”) was widely used long before Jane Austen used it.

In that old sense, information can be useful (e.g., in decision making, in control of actions, or in collaboration with or influencing others) because it involves *reference*, unlike Shannon information, which is primarily concerned only with mathematical features of manipulable, storeable, and transmissable structures, not what those structures refer to.

Not only beliefs and percepts, but also desires, preferences, intentions, evaluations, puzzlement, and other mental states and processes can have information content, often referring to something non-existent but wanted or intended, that is, *control* information, as explained in Reference [8]. This is related to an important distinction between two “*directions of fit*” discussed by Elizabeth Anscombe in Reference [9]. “Belief-like” information states in intelligent individuals need to be changed if they don’t fit the way the world is, whereas if an organism’s “Desire-like” information state does not fit the world, then (if possible) the world needs to be changed, or protected against change, to fit the state—for example satisfying a desire. Desire-like states include intentions, desires, hopes, fears and wishes, in contrast with *beliefs* about past, present or future states, processes, events, and so forth. In the case of desire-like states referring to the past, such as regret or disappointment, things are more complex: actions cannot change the past, though, if correctly informed, they may be able to repair, or compensate for, or take steps to prevent repetition of a misfit.

Many theories of consciousness seem to ignore states whose contents are desire-like in that sense, including desires, intentions, preferences, hopes, fears, wishes, regret, anger, despair, longing, relief, joy, pains, pleasures, and many others. In contrast, great novelists rely on deep, shared, but usually unarticulated, theories about these states and processes.

Most theories of consciousness ignore mathematical consciousness, whose features were partly described by Immanuel Kant in 1781 [6]. Kant is wrongly thought by many (e.g., Hempel [10]) to have been refuted by Einstein and Eddington, whose theoretical and observational research refuted purely Euclidean theories about the nature of physical space-time. What they did not refute were ancient non-trivial discoveries about properties of Euclidean spaces, including Pythagoras’ theorem and many others.

I don’t know of any scientific theory or computational model of consciousness that adequately addresses the enormous variety of states and processes referred to in this paper. I shall attempt to specify some requirements for satisfactory theories and explanations that may lead to new relevant (biological) research, though I do not claim to have produced a satisfactory theory—a task for the future. Perhaps most controversially, I’ll try to demonstrate the importance of many sub-personal information-using mechanisms, and uses of informed control in interactions between separately evolved sub-systems in complex organisms, including our ancient ancestors and many non-human intelligent animals. I suggest that such ancient mechanisms still underpin current human consciousness.

## 2. Ancient Internal Languages

It is generally assumed that humans evolved as spoken language users, with sign languages and written languages as derivative forms of communication. This ignores strong evidence that sign languages must have come first, and internal languages even earlier.

For some reason, very few scientists and philosophers appear to have noticed that insofar as non-human animals, or pre-verbal humans, have percepts, intentions, intentional actions, fears, satisfied desires, and other states with more or less complex *referential* content, they must use internal structured languages with syntax and semantics, to express that content—as I argued in 1978 [11] and 1979 [12]. For example, extending or repairing a partly built or damaged nest requires abilities to perceive structures and relationships and potential for changes, on the basis of which actions are selected and executed. The perceptual processes must produce structured internal states that can engage with other information available to allow actions to be chosen and executed and the results compared with intentions or requirements. So structured useful percepts require internal languages.

A bird whose nest is incomplete or has been damaged needs to take in information about the existing and missing parts and their relationships and derive information about materials and actions required to complete or restore the structure, and use that information to seek and fetch appropriate materials (twigs, leaves, bits of mud, and many others, depending on the type of bird and type of nest) and then insert them in the appropriate location in the appropriate way. For instance a twig that will fall away if merely pressed against a part of an incomplete nest has to be inserted in such a way that it will be held in place by the previously inserted twigs. (How can they get such a process started with the first few twigs?)

Animals that can perceive and distinguish different structured situations and produce actions that create or preserve appropriate relationships must have some means of encoding the information about current states, missing features, and potential actions in a usable form: that is, they must have *internal languages* with suitably rich syntax and semantic expressive power, along with mechanisms for using them. The expressive power required for representing the gap that needs to be filled by fetching a new twig is related to but different from the expressive power required for the constantly changing information used when controlling insertion of a new twig.

The same can be said of a human child that has not learnt to talk but can manipulate its body parts and objects in the environment in order to achieve intentions, such as crawling to a new location, picking up a toy, or putting something in its mouth. In order to have such intentions ahead of actions a child (or robot) must have an internal language (or set of languages) enabling it to deal with, produce, or prevent novel structured configurations of objects, or processes, as a result of perceiving current situations and relating them to possible future situations. Mechanisms are also required that allow the internal language or languages to be put to use, including forming goals, selecting actions, controlling actions, checking results, and also representing goals and possible actions for other individuals. Behaviours with this requirement in human infants and chimpanzees were presented by Warneken and Tomasello in Reference [13], though they did not explicitly mention the representational requirements. Further examples involving pre-verbal children are presented in this discussion note: http://www.cs.bham.ac.uk/research/projects/cogaff/misc/toddler-theorems.html.

The mathematician David Mumford explicitly acknowledges the requirement for pre-verbal humans and other intelligent animals to use internal languages, as explained in Reference [14]. I have not encountered anyone else who has, though perhaps Piaget came close.

These comments apply to all pre-verbal human children who can select and perform actions of varying complexity long before they can use a spoken or signed “external” language. Of course, many of the comments also apply to other species that can perceive and manipulate their environment by performing more or less complex actions—including finding and eating items in the environment, catching prey, escaping predators, finding hiding spaces, creating stores for future use, building nests, destroying nests of competitors, and so on. The ability of an octopus to use a discarded bivalve shell as a portable shelter is a spectacular example, illustrated in https://www.nationalgeographic.com/news/2017/04/coconut-octopus-shell-ocean-video/.

In a subset of species, older members play a crucial role in helping their offspring develop required competences. Such parents need to be conscious of risks and opportunities for their offspring, which requires the use of some form of internal language in which the information is encoded, referred to as “vicarious affordances” (affordances for other individuals) in Reference [15]. In orangutans this close caring goes on for six to eight years, as there is a huge amount for youngsters to learn in dangerous arboreal habitats, where a mistake can lead to a serious fall, producing death or injury. Various forms of other-related meta-consciousness are also needed by intelligent predator or prey animals. This requires internal languages for encoding information about the needs, opportunities, risks and information of other organisms—a form of consciousness normally ignored in models and theories of consciousness. McClelland [16] is an exception.

Evidence that spoken languages must have come relatively late, and were probably preceded by internal languages and sign languages, include the complex modifications of tracts used for breathing, eating and drinking that were required for spoken languages, and the prior availability of highly manipulable hands and brain mechanisms for controlling intricate hand movements for non-linguistic tasks. If much early communication was about actions, then re-using mechanisms for *performing* actions in *communicating about actions* may have been an effective strategy. Moreover, the strong evidence that all normal humans are capable of learning sign languages even if most don’t use them suggests that signed communication may have come first, perhaps as a natural extension of collaborative physical actions (such as helping relatively incompetent young offspring, moving heavy objects, building dwellings, collaborative hunting, sharing dead prey, and other tasks). The switch to spoken languages with newly evolved mechanisms, may have been driven by the need to communicate out of sight, or at night when hands cannot be seen, or while hands are already used for intricate tasks. That switch would have required relatively slow multi-stage evolutionary transitions, including transitions in vocal tracts and brain mechanisms for controlling sound production, discussed in Reference [17].

Infant-toddler-child transitions in humans can be very hard to describe accurately because so little is understood about the brain mechanisms. Very often the research that is done merits the scathing comment by Annette Karmiloff-Smith in Reference [18]: “Decades of developmental research were wasted, in my view, because the focus was entirely on lowering the age at which children could perform a task successfully, without concern for how they processed the information”. Unfortunately, I don’t think the neural net models she (and many other researchers) found attractive have the required computational powers to meet these explanatory requirements. For example, insofar as they express only statistical regularities they cannot be used to derive, or to formulate, discoveries about *impossibilities* and *necessary connections*, one of the requirements Kant had identified for making ancient mathematical discoveries, as I have explained in Reference [19].

The ability to discover that some configurations (collections of relationships) make other relationships impossible, or make other relationships necessary consequences, can have huge effects on pruning search spaces when considering configurations of objects or combinations of actions while trying to achieve some goal or prevent some disaster. So, as I think Kant understood, geometrical or topological mathematical discoveries of the sorts made by ancient mathematicians long before the development of modern logic-based and statistics-based reasoning tools, can have enormous practical importance—being far more powerful than statistical evidence and derived probabilities. Some of those insights are also available to relatively young children: for example *if two solid impermeable rings are linked, no change of location or orientation, or muttering of incantations, will alter the impossibility of separating them*.

I’ll return to consciousness of necessity or impossibility later. I don’t think anyone understands the brain mechanisms that enable the kinds of spatial reasoning that allow impossibilities and necessary consequences to be discovered, using ancient forms of representation that also turned out to be powerful sources of mathematical discovery thousands of years ago, long before mathematicians began to use modern logic and logic-based forms of reasoning. Roger Penrose and Stuart Hameroff, among others, seem to think that quantum mechanisms in microtubules are relevant, but I have never encountered a convincing explanation of how quantum processes in microtubles can produce any particular state of consciousness, such as the consciousness of a squirrel running up a tree-trunk, or a child’s consciousness of a mathematical impossibility, such as the impossibility of separating linked, solid, rings by moving them around in space, or Euclid’s consciousness of the impossibility of a largest prime number.

## 3. Information/Consciousness in Reproduction and Evolution

The roles of information and information mechanisms in biological reproduction and development depend on use of information encoded in DNA in ways that cannot be understood in terms of information states of an agent with beliefs, desires, intentions, and many others. Nevertheless the DNA information, or information in DNA-derivatives, is used in controlling growth, development, maintenance, repair, resistance to infection, and, in some cases, providing services to other organisms in symbiotic relationships. This does not require either the provider or the receiver to have the meta-information that that is what it is doing. Examples include plants providing nourishment (such as nectar or pollen) to insects, and the insects helping to cross-fertilise the plants. In many cases when plants are ready they produce colours or scents that provide information that attracts insects. The insects then help with fertilisation, by transferring pollen from one plant to another, for example. Neither the plants nor the insects have human-like knowledge or intentions, though they provide, and make use of, information. Insofar as they have and use mechanisms for acquiring and using information we can describe them as having primitive forms of consciousness or precursors of consciousness. Many thinkers are reluctant to accept that consciousness can exist without meta-consciousness, that is, without consciousness of consciousness, but that ignores intermediate stages in development and evolution of full human consciousness, including important intermediate stages in development of a fertilised human embryo and important intermediate evolutionary stages.

There are also many non-human examples. Some plants, in soil with impoverished nitrogen content, have mechanisms that detect the arrival of an insect and respond by trapping the insect in fluid and digesting it! The plant need not be aware of what it is doing or why it is doing it. All it needs is awareness (in a minimal sense) of the presence of the insect and the ability to make use of the opportunity. The restriction to the right sort of insect type may be achieved by the evolved design of the trap mechanism rather than use of an explicit recognition process. A subset of the insects must fulfil their needs without being trapped, otherwise neither species could survive. More complex and subtle mechanisms are involved in species where the young derive nourishment from a milk-providing parent, without having to deceive the parent! Evolution has produced a huge variety of cases where willing or unwilling transfer of food, information, or other resources occur, with or without the consciousness thereof in the provider.

There are different uses of information in internal processes once a fertilised egg or seed is in place to develop. Examples include seeds in soil or in fertilized cells surrounded by food in an eggshell, or in a mother’s womb. In all cases, extraordinarily complex processes make use of the DNA, and a flood of diverse chemical products derived from DNA, via RNA, to control development, using information of many different kinds, in various combinations. Although I am far from an expert biologist, it is clear that scientific knowledge about this variety of products and processes has exploded (and fragmented, alas) in recent decades, though there is still much to be learnt about millions of different cases, with many blurred boundaries between stages that may or may not usefully be described as involving consciousness. Blurred boundaries often indicate a need for an enriched ontology/vocabulary.

There are also cases concerned with whole organisms and their abilities to use information in the environment to select and control their actions including a wide range of phenomena discussed in books by James and Eleanor Gibson, for example [20,21,22]. In such contexts it may be more accurate to describe consciousness as being concerned with “what it is like to do X” rather than “What it is like to be X”!

In a more mature science derived from the ideas of Darwin, Huxley, James and many others, it may turn out to be productive to drop the word “consciousness” and instead use a family of new, more precise, related concepts referring to mechanisms, states, processes and forms of (informed) control, involving a generalised notion of use of (biological) “information”. Compare the changing concepts of “weight”, “mass”, “gravitational mass”, and “inertial mass”, in the history of physics. The variety of sub-cases in biology vastly outnumbers the variety of sub-cases physicists normally need to think about, despite all the biological cases being *implemented* in physics/chemistry!

A fertilised egg or seed typically produces a huge variety of physical/chemical structures (molecules or parts of molecules), and physical/chemical processes, while building a new organism. The processes include construction, replication and repair of parts, and also uses of the created parts and materials for many different purposes, such as forming a physical supporting structure (e.g., plant stem, tree trunk, and extensions thereof), with sub-structures for conveying water and other substances either from roots to other parts of the plant, or in the reverse direction, or both! Wherever control is conditional and time-varying (as in online-control) it is likely to merit description as a form of consciousness: active rather than passive, purely receptive, consciousness.

Developmental processes may be wholly or partly controlled by chemical mechanisms derived from the DNA in a new organism (such as a seed or an egg), possibly enhanced at later stages by chemicals absorbed from the environment, including chemical ‘signals’ from conspecifics or other organisms that make use of or make contributions to the organism. Life typically forms a dense network of interacting organisms, except in some extreme environments including deserts with very little life.

Genetic information in a new organism is partly desire-like (in the sense defined above) insofar as it (to some extent) controls processes so that they serve changing needs of a developing individual, though the relationships between information content of DNA and the processes that it influences is typically very complex, indirect and multi-faceted. There are many different *types* of complexity in the processes of gene expression. Very different processes with different forms of control are used in gene expression, such as producing bones, muscles, nerves, outer shells, and other materials in the case of a developing animal embryo, and producing roots and shoots (growing in different directions) in the case of a plant. In contrast, parasitic organisms that mainly use chemical products in their host thereby enormously reduce their requirements for chemical production, and the control decisions they need.

In a more advanced biology we may need to replace concepts like “information”, “consciousness”, and related concepts with a new much richer ontology, to deal with the enormous variety of types of informed control, mechanisms acquiring, using and producing information, and combinations of control in products of evolution, and also recent and future products of human engineering design, with correspondingly diverse concepts related to what we now call “information”. Tibor Ganti [23] has relevant ideas regarding a minimal life form able to maintain and reproduce itself, usefully reviewed/discussed by Korthof http://wasdarwinwrong.com/korthof66.htm.

As organism designs become more complex, developmental choices at particular stages may have consequences for choices at later times. There may be no good options available later if the wrong decisions have been taken at earlier times. But there is not always information available at the earlier time that indicates what will turn out to be best later. One solution to that is splitting an organism into multiple successor organisms with different commitments and needs. Then with luck a significant subset of the choices made earlier will turn out to fit later developments: the new species then continues despite significant wastage of individuals, and diversity in descendents.

An alternative strategy is to develop designs that allow complex choices to be factored into sub-choices that allow decisions to be taken at different times, acquiring information or other resources that can be used in alternative ways later on when the situation has become more definite. Here’s an artificially simplified example. If a shelter is being built that could either be well insulated against future low temperatures, or less well insulated but built more quickly, leaving more time to find food to store, then there may be no way of predicting the future so as to choose whether to add insulation or collect more food. In that case, a possible strategy is to store information about locations where insulating material can be found, and collect food while it is available, and then if low temperatures occur go the sources of insulating material to improve the shelter for low temperatures. In some years the temperature remains high enough for the stored location information to go unused, whereas in other years the temperature falls and insulating material is fetched from known locations and added to the shelter.

Such strategies require the animal to have abilities to seek and store not only food, but also information that can be used later, and abilities to detect when it will be worth using that information. This can easily be generalised in various ways, including use of information about information extending varieties of consciousness.

Situations can arise where there are multiple possible future needs and multiple possibly useful sources of information, with interactions between possible actions at choice points. For example, there may be two sources of materials that meet need N1 and one of them is further away but close to a source of materials that will meet need N2. Then if need N1 arises, choosing which source to go to can depend on whether need N2 has also arisen, or whether there is evidence that it is likely to arise. This is a very simple example of a tradeoff between complexity of information processing and ability to cope with challenging situations. I take it as obvious that it is also an example of (primitive) decision making—a primitive form of consciousness.

So, even simple organisms can benefit from increasing complexity of use of information. There has been a vast amount of empirical research and theory construction in biology, along with many practical applications, including including understanding complex pathologies and finding new types of prevention or medical treatment. But this research barely scratches the surface of the complexity waiting to be understood, and perhaps put to use, in future. Whether something like our current concept (or concepts) of “consciousness” will remain useful in science and engineering, or endure the fate of “phlogiston”—i.e., made redundant by an alternative more powerful collection of concepts—remains to be seen.

One feature of biological evolution that seems to be of great explanatory importance is the tendency of evolution to produce a huge and steadily growing collection of “biological construction kits” for building novel materials, novel structures, novel combinations of solutions to problems, and novel construction-kits! An interim progress report on this idea was published in Reference [24] though more recent developments in the theory can be found here http://www.cs.bham.ac.uk/research/projects/cogaff/misc/construction-kits.html.

One of the recent side-shoots of that research is the investigation of varieties of compositionality in evolution, begun in this document: http://www.cs.bham.ac.uk/research/projects/cogaff/misc/compositionality.html. This, in combination with the theory of construction kits, seems to promise new insights into forms of complexity and variety in evolution and development, including complexity and variety of both biological and artificial information processing (biological and artificial consciousness?).

## 4. The Importance of Quantum Mechanisms for Biology

The biological processes we are investigating depend on non-Newtonian physics because, as Schrödinger argued in 1944 [25], only features of quantum physics can explain the combination of mechanisms allowing reliable long term storage, reliable copying in reproduction and rare but not impossible structural transitions. His analysis was part of the background to the discovery of the double-helix structure of DNA and the steadily increasing complexity of mechanisms found to be involved in processes through which DNA sub-structures, or their derivatives, including RNA and its derivatives, influence structures, mechanisms, functions, and processes, not only during reproduction but also at many stages of individual development and in many aspects of physical functioning in an organism, for example use of digestive systems that decompose food into much simpler chemical structures and make appropriate use of the resulting chemicals in building new structures, enlarging old structures, replenishing energy stores, sending required chemicals to damaged sites and using them for repairing tissue damage.

Another type of use is synthesis of new specialised chemicals required for destroying or disabling invading organisms or helping to getting rid of waste products—transporting them to waste outlets in the body, including the outer skin, where that is permeable.

Insofar as such processes depend on detection of needs, opportunities and constraints and selection between alternative responses or alternative levels of response, they can be thought of as making use of *minimal forms of consciousness at molecular scales*. Some forms of consciousness, for example awareness of risk of unwise responses to phone calls from unknown persons that can lead to financial fraud, require a whole functioning human mind. In contrast, crucial parts of an organism can have forms of consciousness that the whole organism lacks, for example detection of a chemical needed to support repair of minor physical damage, or use of pressure sensors in feet or vestibular signals, to control muscular tensions to maintain balance. There are many processes of which we are normally unconscious but whose effects become conscious when they malfunction. But their normal functioning involves use of information: a form of low-level consciousness.

Physical/chemical structures and processes are involved in development of both relatively stable *parts* (on many scales) of individual organisms, and also many forms of *process control*, including: control of production of new physical parts during development, maintenance of those parts (including damage detection and repair and immune reactions), and highly complex coordinated uses of collections of body parts—collections whose members can change dramatically during individual development, including changing uses of a multitude of muscles required for locomotion, or for linguistic communication, including in some cases switching from spoken to sign languages in communities with a high rate of congenital deafness.

The portions of DNA provided in the initial cell and the many copies made later do not simply provide a fixed specification of a new individual, since copying and use of fragments of DNA or derived molecules is subject to an enormous number of variations, depending on context. This can be thought of as construction of life forms partly on the basis of cooperation and interaction among an enormous variety of fragments of consciousness, implicitly orchestrated by a network of detected needs and opportunities. In animals this often involves seasonal changes, including mating behaviours and nurturing behaviours, for example. In many plant species, much of the influence of information derived from DNA is also cyclic, including dramatic seasonal changes (such as growing, then shedding massive numbers of leaves), along with a host of relationships with other organisms. Some of those are symbiotic relationships and uses of other species in fertilization, reproduction, and dispersion.

There is a growing community of researchers who describe themselves as studying *plant consciousness*, including types of acquisition and use of information by plants. Similar comments can be made about research on consciousness in insects, microbes, and other information-using organisms that are not normally included in philosophical discussions of consciousness.

Very complex and still only partly understood forms of gene expression in humans make possible picking up, using, and contributing to changes in, one or more human languages, including spoken languages, written languages, sign languages, communicative facial expressions, body postures, mathematical notations, and in recent decades, production of a huge variety of programming languages with very different syntactic, semantic, and pragmatic features.

Some of the results of these mechanisms are easily observable structures and behaviours—such as the changing shapes and behaviours of a complex growing animal, including display of sexual motivations, preferences, behaviours and physical features during puberty—while other results of developmental processes are sub-microscopic molecular control processes including distribution and use of products of digestion, tissue growth and repair, and various kinds of learning achieved by immune mechanisms. Insofar as these control processes do not merely transmit physical influences through physical mechanisms (as levers, pulleys and gear wheels do) but acquire and combine sources of information in context sensitive ways, we can describe them as more or less *primitive* forms of consciousness.

We’ll see later that “a meta-configured genome” can lead to dramatic changes in some of these competences and behaviours across generations, including understanding and use of increasingly complex mathematical discoveries.

In contrast, the extended behaviours of rivers, masses of snow on mountains, moving continents, volcanoes, are not normally taken to be examples of life or consciousness for reasons that are left as an exercise for readers!

In short, consciousness in living organisms involves, on the one hand, *reception* and *storage* of information and, on the other hand, complex, active, co-ordinated, *use* of that information on many space-scales and time-scales, influenced by many different factors, including skills, needs, wants, resources, dangers and other background information. I shall try to spell out some of the implications of these ideas, including the possibility of a highly ambitious research project on varieties of evolved, or evolvable types of consciousness, including mathematical consciousness.

## 5. Philosophical Over-Confidence and Psychological Myopia

Unfortunately, many philosophers (but not only philosophers) seem to think they know from first-hand experience of consciousness all there is to know about what needs to be explained, so they ignore most of the important variation in contents, functions, and mechanisms of consciousness, both over evolutionary time scales and during individual development. (I sometimes get this impression when non-philosophers, or beginner philosophers, echo a “what it’s like” mantra when discussing consciousness.)

Even outstanding work by psychologists, presenting rich and varied collections of features of consciousness (usually visual consciousness, as in Frisby’s [5]) can be too narrow because most of the *functions* of consciousness (including details of visual or spatial consciousness) are ignored, including the continual mundane uses of consciousness in interacting with the environment to avoid risks or to acquire resources and satisfy needs.

Most such work ignores the types of consciousness involved in ancient mathematical discoveries in geometry and topology, many of which are related to uses of spatial consciousness in selecting and performing actions, making and using tools, creative engineering or artistic design, collaborating or competing with other individuals, interacting with prey or predators, or helping active developing youngsters whose abilities and needs continually change with time. A messy, incomplete, and still disorganised collection of examples from observation of young children was mentioned earlier in Section 2
http://www.cs.bham.ac.uk/research/projects/cogaff/misc/toddler-theorems.html.

Some psychologists make the mistake of thinking that reliably repeatable laboratory experiments and observations, possibly enhanced using brain-scanning technology, can answer all questions, thereby ignoring most of the evolutionary and developmental history, and the mostly sub-microscopic (especially sub-neural, and more generally sub-cellular) mechanisms and processes that are difficult or impossible to observe directly. Grant [26] and Trettenbrein [27] are among the neuroscientists who now counter this trend. The variety and depth of investigations into sub-neural mechanisms that challenge widely-held assumptions about consciousness and related phenomena (including action control and ontology extension) may expand enormously in future, partly because our understanding of questions to ask increases and partly because our technology for detecting, observing, measuring and interacting with sub-microscopic processes increases.

Those who rightly emphasise the relevance of biological evolution can also make mistakes, such as failing to notice that biological evolution necessarily involves *discrete* transitions since there can be at most a finite sequence of parent-offspring steps between any ancestor and any descendant, except in the case of vegetative reproduction; though even that typically involves discrete cell production. This suggests that a key sub-goal of our investigation should be to identify important *discontinuities* (both during evolution of consciousness, and during individual development) relevant to changes in contents, functions and mechanisms of consciousness.

I tried to specify some of the requirements for adequate explanations of human forms of consciousness, including mathematical consciousness, in an earlier paper [28], which includes examples that will not be repeated here. This new paper presents new details, emphasising evolutionary processes, partly triggered by presentations at the 2019 *Models of Consciousness* conference in Oxford. In particular, I have tried to extend some old ideas about evolution of consciousness by Darwin, Huxley, William James, and others mentioned above, and some recent extensions to the *Meta-Configured Genome* theory summarised in my Oxford presentation https://www.youtube.com/watch?v=0DTYh37U8uE.

We need to avoid two common mistakes, one made by philosophers (many of them influenced by Nagel [29]) who think that their own experience in moments or states of consciousness (“what it’s like…”) fully determines what needs to be explained, ignoring most of the diversity of types of consciousness. For instance, I know of none who have tried to apply that approach to ancient kinds of mathematical consciousness leading up to the great mathematical discoveries of mathematicians such as Archimedes, Euclid and Zeno.

Another common mistake leading to impoverished theories is the assumption that evolutionary trajectories form a branching *tree-structure* with a common root, or possibly disjoint trees with different roots; whereas the evolutionary topology that arises from various types of symbiosis, sexual reproduction, and a meta-configured genome (summarised below) is far more complex than any tree structure. This is illustrated by the well known cross-language variation in linguistic competences at all linguistic levels, from basic sounds or signs to construction of phrases, sentences and paragraphs, all generated by an (approximately) common human genome in different cultural contexts, producing several thousand different human languages, without any initial help from teachers of those languages, who did not exist in advance and therefore could not have taught the new languages.

No current theory that I know of accommodates or explains most of the detailed variations in human consciousness based on a (mostly) shared human genome interacting in complex ways with very varied and increasingly complex environments, including physical and cultural environments—produced in part by products of that genome. This paper is a partial progress report on a very ambitious project, the (Turing-inspired) Meta-Morphogenesis project, sharing some of the presuppositions of James, in Reference [1], but aiming to produce a more complete theory characterising some of the often ignored variety of products of evolution, including increasingly complex forms of compositionality (summarised in http://www.cs.bham.ac.uk/research/projects/cogaff/misc/compositionality.html).

An enormous, but still growing collection of evolved biological construction kits constantly maintains and enriches that diversity, including construction kits for building various physical structures, various information processing mechanisms (starting with simple control mechanisms) and also construction kits for building new construction kits, as explained in http://www.cs.bham.ac.uk/research/projects/cogaff/misc/construction-kits.html.

Much of this work was inspired by reflecting on Turing’s motivation for his 1952 paper on morphogenesis [30], which prompted the question: why was he writing about chemistry-based morphogenesis so soon after his more widely known 1950 paper [31], the only one usually cited by consciousness theorists? There is a clue in one sentence of the 1950 paper: “In the nervous system chemical phenomena are at least as important as electrical”, which is rarely mentioned by commentators. Thinking about that sentence at the time of Turing’s centenary triggered the birth of the Meta-Morphogenesis project, which has since grown considerably: http://www.cs.bham.ac.uk/research/projects/cogaff/misc/meta-morphogenesis.html.

## 6. Varieties of Biological Consciousness

So far, I have emphasised increasing variety in evolved biological mechanisms, processes and their products. In what follows I shall present some differences in types of biological consciousness, in order to illustrate some of the individual and cultural variation, as well as cross-species variation, predicted/explained in the Meta-Configured Genome theory (MCG) summarised below. Different types of consciousness will require different explanatory mechanisms operating in different environments with different individual developmental histories—for example the differences between types of consciousness in honey bees and the consciousness in a biologist studying bee consciousness https://www.scientificamerican.com/article/exploring-consciousness/.

What all organisms (including sub-microscopic organisms invading other organisms) have in common is acquisition and use of semantic information (called ’Austen information’ in Section 1), some of it about their own body state and current needs (including food, rest, or reduction of discomfort, based on internal sensors) and in some cases semantic information about the *environment* acquired through sensors, some of which provide information about where various opportunities, resources and dangers are, including where other organisms are and what they are doing, whether their presence is a threat or not, and so on. The sensing/perception mechanisms involved are much more complex and varied than assumed in most discussions of consciousness by current philosophers and scientists.

The information produced and used to attract mates and the camouflage information produced to deceive predators or prey differ in the detailed mechanisms and in their effects on behaviour, but both illustrate the same general points. Moreover, information can be intentionally mutually shared across individuals of different species, as when a child and a dog play with each other, with clear indications of “mutual metacognition”, including a dog repeatedly bringing a stick to be thrown by the child.

Sophisticated mechanisms of self-monitoring have been produced in human evolutionary history, and enhanced by various locally developed uses of those mechanisms, some taught explicitly, for example in training for meditation or acting, but most taught implicitly through explicit or implicit teaching of a huge variety of skills, including linguistic skills, artistic skills, and practical skills used in designing, making and repairing things. All provide opportunities for improvement through self-monitoring, including detecting previously unnoticed sources of error or omission, discovering untried options, or noticing puzzles that lead to new insights, such as “What do I need to do to avoid that sort of mistake in future?” “How did I get that effect, and how can I repeat it, or prevent it?” “Is there a quicker or more elegant way of solving this problem?” “How can I help others understand what I am doing?” These all obviously involve aspects of consciousness, including self-consciousness. Unfortunately most theories and models of consciousness focus only, or mainly, on consciousness of perceived facts (how things are or appear to be) ignoring consciousness of problems and gaps in information.

Information that is not produced or used intentionally, includes the *genetic* information that (partly) controls physical development and development of information processing capabilities in individuals. One of the most powerful “discoveries” of biological evolution is that if a genome does not specify all the details of an organism, but instead allows some details to be influenced by information gained from the environment encountered during development, this can allow a species to flourish in a wide range of different environments—illustrated in relation to plants by Heslop-Harrison [32]. In such a case the potential variety is implicitly specified in the genome but individual-environment interaction produces a selection from that potential. Meta-configured genomes significantly extend that flexibility.

## 7. Meta-Configured Genomes

A more powerful evolutionary invention was what Jackie Chappell and I call a Meta-Configured Genome (MCG), in which variability of end-products (adult forms) is considerably increased by allowing later stages of gene expression to use a level of abstraction that permits generic specifications to be instantiated within an individual by using “parameters” acquired through exploratory behaviours triggered at earlier stages of development. If the environments encountered at earlier stages are themselves deeply influenced by older members of the species (including products of long dead ancestors!) this can give new individuals abilities that none of their ancestors had, because their ancestors did not encounter the required influences during early development. In other words, the ancestors’ parametrised multi-layer genome lacked some of the *inputs* to gene expression that descendents have as a result of the achievements of those ancestors.

A recent striking example of this is the production of new electronic, especially computational, devices, allowing young humans to encounter such devices at a much earlier age than their parents did (if their parents did), and thereby enabling young individuals to develop new layers of competence that build on information gained in those early encounters. Such multi-layer “feedback” from adults to early gene expression in their offspring can produce very different effects in different sub-cultures. One of the mechanisms driving such processes is a motive-generator that detects opportunities for triggering new motives which, if acted on, may provide information that will be useful later, though the individual cannot know in advance what the information will be or how it will be useful. I call that “architecture-based” motivation (ABM), in contrast with “reward-based” motivation, a distinction that is explained in this document (which updates an earlier published version): http://www.cs.bham.ac.uk/research/projects/cogaff/misc/architecture-based-motivation.html.

A much older and more spectacular illustration of MCG in operation is the fact that the human genome allows processes of development of layers of linguistic competence to acquire parameters from the environment that help to shape *later* linguistic development in novel ways. As a result, humans do not *learn* languages, they *create* them cooperatively. A famous example of this occurred when deaf children in Nicaragua cooperatively developed a new sign language because their (deaf) teacher had been taught to sign relatively late, because until then the national policy had been to teach deaf children to lipread, on the mistaken assumption that that would enable them to communicate better with the non-deaf population. As a result the teacher had learnt only a relatively simple sign language, that did not meet the needs of the young deaf learners. They therefore collaboratively created a new sign-language, as demonstrated in this video report https://www.youtube.com/watch?v=pjtioIFuNf8, apparently a spectacular example of the MCG mechanism at work. This and other examples are discussed in more detail by Senghas in Reference [33].

Compare the remarks in Section 2 above about linguistic development. The fact that co-operative language *creation* rather than language *learning* is going on in young children is normally obscured by the fact that, unlike the Nicaraguan deaf children, most young humans are in a tiny minority of language developers, where the majority have already made major selections between alternative options. However, twins sometimes exhibit cooperative development of a new private language, a phenomenon known as “Cryptophasia”, though I do not think the genetic basis for such creation described here has been generally recognised.

These ideas are still far too schematic. There are very many different types of learning or development that seem to use something like the pattern described here, but many more cases need to be described, with detailed analyses of the various types of Meta-configured mechanism and their biological uses, including roles in mathematical discovery and other forms of consciousness. Research is also needed on the biochemical mechanisms in reproduction that underpin Meta-Configured genomes.

In particular, if the MCG theory is correct in suggesting that a late developing aspect of a human genome can form patterns of competence (including linguistic or mathematical competence) that are instantiated using information previously acquired through behaviours triggered for the purpose by special forms of motivation at earlier stages of development, then an important research problem is to identify the physical mechanisms involved. If, as suggested by Penrose and Hameroff at the Consciousness conference and elsewhere, sub-neural microtubules are involved in consciousness, it should be possible to explain how the information from different sources (genetic and previously picked up via specially motivated actions) is combined to produce the new forms of consciousness, for example using previously picked up information fragments and a newly instantiated grammatical schema to produce a new insightful thought or utterance, involving a powerful new (to the learner). conjecture, question, or explanation. At present it is not clear to me how microtubules could contribute. If they do, that must involve developmental processes that have so far not been discovered.

## 8. Evolved Networks, Not Trees

Taking into account features of sexual reproduction, symbiosis, and structured individual development, implies that evolutionary *branching trees* need to be replaced by more complex *tangled networks*, as in the theory of *symbiogenesis*, promoted by Lynn Margulis and others [34]. This implies that forms of biological consciousness that combine different evolutionary heritages may have different functions and mechanisms, and instead of forming a branching *tree* of types they are likely to form a *network* of types without a unique ordering relation.

The Meta-Configured Genome (MCG) theory [35] (still under development here http://www.cs.bham.ac.uk/research/projects/cogaff/movies/meta-config/), implies that proposed *unitary* data-driven mechanisms of consciousness are biologically inaccurate. Instead, different abstractly specified mechanisms are used at different stages of gene-expression—different stages of individual development—often using parameters acquired during earlier stages, which can vary under the influence of geographic, or cultural differences, and also differences in individual developmental histories—including, in some cases, the need to cope with severe physical abnormalities.

The diversity and power of biological evolution and its products are enhanced by the use of multi-layered, temporally staggered, gene expression, where later stages of gene expression use *generic structures* that are instantiated using environmental parameters acquired during earlier stages. Because of this, a meta-configured genome allows later stages of gene expression to have very different effects in individuals with different environmentally influenced developmental histories, including allowing young learners to make developmental transitions that were *impossible* for their parents and ancestors. This is illustrated by the huge diversity among human languages and other competences resulting not from genetic diversity but from diversity of cultural heritages within which genes are expressed—as predicted/explained by the MCG theory. This applies also to the diversity of human technology, science, social practices, forms of education, religions, art and value-systems. (These claims should ideally be supported by examples and evidence that would fill many books.)

This is obviously true for language development in humans, in view of the multi-layered diversity of human languages, with differences at every level of structure, each level based partly on some genetic influences, often using delayed gene-expression, and partly on earlier environmental influences, for example local linguistic usage and locally developed technologies. An individual’s language may also have individual features or quirks that come from unique genetic features of the individual’s brain or physiology, or unique, inexplicable, twists in the individual’s development. Occasionally human twins develop a private language based on “opportunistic” mutually supported gene-expression.

Theories that claim that all human languages are essentially the same ignore all these facts. Such sources of uniqueness can also affect other competences, including musical, artistic or mathematical talents.

Similarly, consciousness can have different contents, functions, and mechanisms at different stages of development, where each stage combines abstract structures influenced by newly expressed genetic information, with information acquired from the environment or from internal sources at earlier stages of development. Because children can make advances not made by their parents, they can change the environment in which the following generation of children develop, that allow the new children to develop competences none of their ancestors had, and the same can happen to their offspring. That is the MCG (Meta-Configured Genome) thesis summarised in my Oxford MOC talk, mentioned above.

Some of the mechanisms in intelligent species (including squirrels, crows, apes, elephants, etc.), provide abilities to detect and make use of *impossibilities* (such as: a rigid object with minimum diameter larger than a fixed width gap cannot pass through the gap in any orientation) and *necessities* (e.g., containment is necessarily transitive).

As Immanuel Kant noted (in Reference [6]), these insights cannot be based entirely on empirical generalisation, or on logical deductions from definitions. Other mechanisms are required—though he thought humans may not be clever enough to discover them—suggesting that such knowledge may depend on “an art concealed in the depths of the human soul”. I suspect that if Kant had learnt to write computer programs, he might have tried to build a working model of that “art”.

Some of these ideas also appear in claims by Roger Penrose, one of the invited speakers at the MOC conference. He and I use partly overlapping geometrical examples—in my case influenced originally by Kant—but so far I have not understood how his and Hameroff’s (microtubule-based) mechanisms can explain the ancient examples of mathematical insight into spatial *necessity* or *impossibility*. Some philosophers of mathematics believe that only the use of logic-based forms of representation and reasoning can do that. But that leaves unexplained the abilities of ancient mathematicians and intelligent non-humans without access to the notations and techniques of modern logic. I suspect that a long term complete solution will have to make use of much deeper theories than we currently have about the evolution of varieties of information processing, in humans and other intelligent species, and the varieties of individual developmental mechanisms so far produced by combinations of biological and social/cultural evolution. Producing researchers equipped to advance this research may require substantial extensions to our educational practices.

## 9. The Need for “Other-Directed” Kinds of Metacognitive Creativity

The ability to *detect* and *use* spatial/mathematical (including geometric and topological) possibilities, impossibilities and necessities, which we seem to share with some other intelligent species (including apes, squirrels, and crows) does not necessarily provide abilities to *reflect on*, *reason about*, *communicate* or *teach* what has been learnt. For that, additional evolutionary/developmental processes are required, extending *self-directed* meta-cognitive mechanisms so as to include *other-directed* meta-cognition that can be used by good educators (including attentive parents) to pass on what they have learnt, enabling later generations to acquire important knowledge more easily and at an earlier age.

In great human teachers, additional meta-cognitive mechanisms provide reflective mathematical consciousness (a kind of self-teaching ability) that can repeatedly lead to new mathematical discoveries, as seems to have happened in ancient geometers such as, Appolonius, Archimedes, Euclid, Zeno, and many others.

Unfortunately, in the UK and many other countries, most schools no longer teach Euclidean geometry, including ancient ways of finding constructions and proofs. (Some Eastern European countries seem to be exceptions.) So most graduates, including mathematics, science, engineering, and philosophy graduates, now fail to understand most of the topics I have been describing. (Some readers may find the mixed, slowly growing, collection of examples here useful: http://www.cs.bham.ac.uk/research/projects/cogaff/misc/impossible.html. However, I do not believe any currently proposed mechanism in psychology, neuroscience, philosophy or AI is capable of explaining these ancient abilities to make deep new mathematical discoveries.

In 1936, Turing proposed, in Reference [36] a distinction between *mathematical ingenuity* and *mathematical intuition*, that seems to be related to this. He claimed that computers could replicate human mathematical *ingenuity* but not mathematical *intuition*, but he did not say why. I conjecture that he had the same insight as led Kant to talk about some mathematical discoveries as being *non-empirical*, *synthetic* (non-analytic, i.e., not based solely on definitions and logical reasoning) and *necessarily true*. I suspect (and I think Turing suspected) that the mechanisms required cannot be provided either by current, digital, logic-based forms of computation, or by neural statistics-based learning mechanisms. This may have motivated his 1952 paper on chemistry-based morphogenesis [30] (usefully summarised for non-mathematicians by Ball in 2015 [37]).

Could the explanation for that difference between human mathematical competences and what digital computers can do lie in the use by human brains of sub-neural molecular computational mechanisms with their combinations of continuous and discrete processes, emphasised by Schrödinger in 1944 in *What is life?* [25]? If so that has serious implications concerning limitations of currently fashionable AI theories and mechanisms. This suggestion is motivated by the fact that the space of logical/algebraic formulae and proofs and operations thereon, to which most current professional mathematical thinking seems to be constrained, is very different from the space of continuously deformable configurations of lines, circles and other shapes that seem to have been the focus of much ancient mathematical thinking—and was the focus of my thinking about geometry as a student in the 1950s, though very few mathematics students seem to encounter that space nowadays.

Perhaps Turing’s experience of using continuously deformable diagrams during his mathematical education explains why he wrote in his 1950 paper: “In the nervous system chemical phenomena are at least as important as electrical”. Unfortunately he gave no explanation. Perhaps, as hinted by Newport in Reference [38], von Neumann had similar thoughts towards the end of writing his unfinished, posthumously published book written in 1958 [39]. I do not know whether he had read Turing’s 1952 paper.

Working out these ideas in more detail may require younger, more competent, brains than mine! Perhaps this framework will motivate and help them investigate possible explanatory mechanisms in sub-neural chemistry. As stated previously, I do not know whether the ideas regarding microtubules presented by Stuart Hameroff at the Models of Consciousness conference can provide the missing mechanisms. I don’t see how they could explain ancient geometrical discoveries.

Many more detailed discussions and illustrations are linked from the online website for the Meta-Morphogenesis project, under continuing development. http://www.cs.bham.ac.uk/research/projects/cogaff/misc/meta-morphogenesis.html. Collaborators would be very welcome, including researchers working on the many varieties of consciousness in non-human organisms, and the explanatory powers of sub-neural forms of chemical computation, discussed in References [26,27], among others.

## 10. Mathematical Cognition Supporting Kant

Around 1959, most of the philosophers I encountered made claims about mathematical cognition typified by Hempel’s arguments [10] against Kant’s philosophy of mathematics. But they all seemed to ignore personal experience of *doing* mathematics, especially finding constructions and proofs in Euclidean geometry. So I obtained permission to switch from mathematics to philosophy, and completed a thesis defending Kant, in 1962 [40]. Two papers based on the thesis were published in 1965 [19,41], before I encountered AI (in 1969) and started learning to program, hoping to build a (physical or simulated) robot with spatial reasoning capabilities that could also support ancient forms of mathematical discovery. Some ideas about possible ways of making progress were presented in my 1978 book [42], but but the project proved very difficult. At the time of the Turing centenary (2012) I began to wonder whether digital forms of computation were not up to the task, and suspected that that was why Turing, sixty years earlier in Reference [30], had investigated chemistry-based mechanisms, combining continuous and discrete processes, as mentioned in Section 9, above—that is, possibly because chemistry-based reasoning mechanisms are richer than logical reasoners, since chemistry provides useful combinations of discrete and continuous processes (which also make those mechanisms useful in biological reproduction, as noted by Schrödinger, cited above). However, I do not know anyone who is able, to present a theory about how brains might use chemical information processing to discover and prove the ancient theorems of Euclidean geometry. Perhaps the eventual explanation will refer to the ability of chemical computations to determine whether a certain type of molecular structure can be derived from other molecular structures, by transferring appropriate sub-particles (protons, electrons, etc.).

That will require mechanisms demonstrating how various kinds of reasoning in continuous spaces are possible, including, for example, demonstrating that it is possible to have three simple closed curves C1, C2, C3 on the surface of a torus where C1 can be continuously deformed into C2 and vice versa, but neither C1 nor C2 can be continuously deformed into C3. Readers should not find it hard to imagine instances of such curves on a torus. Discovering that C1 can be deformed into C2 on the surface, and *vice versa* may be relatively easy, by trying to imagine the deformations. Reasoning in a different case that C1 and C2 *cannot* be continuously deformed into each other is typically much harder because you somehow have to survey all possible continuous deformation processes changing C1, and somehow work out that none of them can produce the shape of C2. How can a brain perceive such impossibility? It requires an ability somehow to consider an infinite collection of possible curves, such as all the possible results of continuously deforming C1 on the surface, and check that none of those possibilities coincides with C2. How can that be done, either by a brain or by a computer? We know that at least some brains can do it (including brains of many mathematicians with geometric and topological reasoning expertise), but not how they do it.

There are much easier cases: for example if P, Q and R have Boolean values, that is, each is either true or false, then it is easy to get a computer to check whether a boolean expression involving P, Q and R, such as **P & Q & not-R** is capable of being True, because each of the three variables must be either True or False, so there are only 8 possible combinations and a computer can easily be programmed to examine all of them using a finite truth-table. But many geometric or topological examples, like the questions above about continuous deformations, require abilities to reason about infinite spaces. If two curves each connect two points on a continuous 2D surface, then if one of the curves can be continuously deformed into the other there will normally be infinitely many ways of doing that. Likewise, if some task is to be proved impossible that will often require investigating infinitely many possible transformations to show that none of them can perform the task.

In 1899 David Hilbert published the German version of his [43] and demonstrated that all the theorems of Euclid’s *Elements* could be derived from a finite collection of axioms using standard methods of logical inference. But it is arguable (and Gottlob Frege did argue) that Hilbert had, in effect, changed the meanings of the terms used by Euclid, such as *point*, *line*, *intersects*, and so forth, since for Euclid they referred to spatial structures and relationships and we can ask whether the statements about spatial structures and relationships made by the axioms or theorems are true or false, whereas for Hilbert all that matters is that every axiom or theorem in Euclid corresponds to a unique formula in Hilbert’s system, and all the theorems in Hilbert’s system can be shown to be logically derivable from the axioms in his system.

Against the claim that Hilbert has shown that Geometry is reducible to logic in the same way as Frege (and others) had shown that Arithmetic is reducible to logic, it can be argued that even if there is a subset of logic (defined by a collection of axioms and definitions for numbers and operations on numbers) that in some sense is isomorphic with arithmetic, the *original* discoveries in arithmetic made by ancient mathematicians were not simply discoveries about certain logical formulae being derivable from other logical formulae plus definitions, but were discoveries about properties of collections of discrete objects and properties of counting operations and one-to-one correspondences (bijections). For example, understanding the many practical applications of number concepts requires grasping the fact that the one-to-one correspondence relation is necessarily *transitive*, as mentioned in Chapter 8 of Reference [42]. Then the discovery of the existence of prime numbers involves discovering that there are some collections of objects that cannot be rearranged to form an array of N rows and M columns where both N and M are larger than 1.

A child playing with buttons on a flat surface divided into squares may try to rearrange rows of eight, nine, ten and eleven buttons into rectangular form, succeed on the first three and fail with eleven. What mechanisms are available to brains at different stages of development to attempt to explain why the failure occurs? At present, I do not think anyone knows. Another example: Euclid’s discovery that there is no largest prime number is not just a trivial logical consequence of some definitions but has an enormously rich collection of consequences, including consequences that have been found useful for security systems. I hope these remarks convince readers that so far nobody has accomplished the task of explaining how brains make possible the forms of mathematical consciousness required for understanding and using cardinal and ordinal numbers in everyday life.

## 11. Conclusions

This paper cannot easily have a satisfying concluding section, as more questions have been asked than answered, and it seems clear that the collection of examples of different sorts of consciousness presented here merely amounts to a few scratches on a huge surface waiting to be carved up more systematically.

I have tried to indicate some of the huge unnoticed variety of types of sub-personal consciousness that need to be explained in a complete theory of how currently known minds work and how many importantly different evolutionary stages and types of consciousness there must have been between the earliest biological life forms, and modern animals, including humans.

I am not the only researcher trying to understand evolved varieties of consciousness, and I cannot claim to have encountered and studied all the relevant publications in the field. However, I find that publications on consciousness written by philosophers, psychologists, neuroscientists and AI researchers tend to fail to address many of the relevant biological phenomena, including sub-organism varieties of consciousness, and cases of conscious performance of difficult or risky or enjoyable *tasks*, as well as conscious *passive* experiencing. Two papers by Prentner that I encountered only very recently also seem to be attempting a broader than normal approach, though I have not yet studied them in sufficient detail to comment [44,45].

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
