# Peer review of "Varieties Of Evolved Forms Of Consciousness, Including Mathematical Consciousness"

_entropy, 2020, doi:10.3390/e22060615_

Round 1

Reviewer 1 Report

Very interesting and relevant. Chemical level of thinking/consciousness has been neglected, especially in AI circles. It is worth bringing forth, which the article is doing well. Also great points about multi-layered genome. I wish the author drew clearer, more direct, implications for AI. The author declares "If this paper is accepted I’ll try to extended it with more detailed suggestions for future research, and a more detailed account of some ideas about multi-layer information processing architectures produced by evolution". I strongly recommend that the paper should be accepted and extended in accordance with the above note.

One philosophical infelicity: T. Nagel is mostly known for his 'what it is like to be a bat' paper; but this is mostly since the paper is easy to teach in Philosophy 101. In the same book The Mortal Questions Nagel ends with a much more substantial article 'Subjective and Objective'. It is further developed in his followong book The View from Nowhere. Nagel's position there is based on Kant's transcendental metaphysics. Some analytical philosophers do not understand Kant's metaphysics these days, but since the author of this article does understand the first Critique, Nagel's project would be refreshing (though Nagel rejected it in his embarassing 2012 book). 

Sloman's broad-brush article broadens the horizon in information theory, genetics, AI, but also philosophy..

Author Response

First I would like to thank the reviewer for proposing that I be given a chance to submit an extended version of the paper. The version I submitted was essentially incomplete insofar as I had over-committed myself in the preceding weeks. Progress preparing a revised version has been slow mainly because I was recovering from a hip replacement operation on 17th March and this has restricted my time spent at a computer, but also because I received some important news recently that is very relevant to this paper, and which required immediate action. I am not permitted to give details at present.

I thank the reviewer for drawing my attention to aspects of Nagel's work that I had not encountered. It does not seem to be centrally relevant to  my paper, so I'll postpone finding out more until after I have finished the paper. A small comment is that many who write about consciousness and the so called 'hard problem' seem to think that the discussion all started in the second half of the 20th century, whereas I think recognition of the depth and difficulty of the problem of accounting for the existence of consciousness in a physical world is much older and was famously highlighted as a problem for Darwin's theory of evolution by one of his strongest supporters T.H.Huxley. The problem was also clearly identified by William James in 1879 as cited in my paper. Moreover, I think James also provided important hints as to how the problem might be solved by a detailed account of evolutionary transitions producing increasingly sophisticated forms of information processing in organisms. Unfortunately, I think most of the recent writing by philosophers, neuroscientists, psychologists and others about consciousness ignores those important evolutionary considerations and their implications.

The reviewer writes: "The author declares "If this paper is accepted I’ll try to extended it with more detailed suggestions for future research, and a more detailed account of some ideas about multi-layer information processing architectures produced by evolution". I strongly recommend that the paper should be accepted and extended in accordance with the above note."

I have now submitted a revised version that is very much longer than the original and attempts to show how we can pursue a broader, deeper study of consciousness than is common nowadays by going back to 19th century recognition of the problem of showing how the phenomena of consciousness can be among the products of biological evolution as a scientific task as well as a philosophical task. I'll add details that those early thinkers could not have known about which have come from more recent scientific work on biological evolution and the many forms of development of living organisms.

The reviewer writes: "Sloman's broad-brush article broadens the horizon in information theory, genetics, AI, but also philosophy.." That is exactly what I am trying to do though I am painfully aware that it is a huge project and I need to find a way to attract a variety of collaborators with very different backgrounds, including biologists familiar with the most recent developments in biochemical processes in gene expression and in the functioning of products of gene expression.

I regret that I ran out of time to follow the link the reviewer gave to some of Nagel's work that I have not encountered, but I suspect, from the reviewer's description, that it would not have contributed to the scientific project I have been trying to specify.

Reviewer 2 Report

One of the main themes of the article is that consciousness is a complex and heterogeneous phaenomenon, and that usual approaches (both from philosophy and from cognitive science) often ignore such complexity.

It is a reasonable claim, with which I am personally sympathetic.  However, the article is rather intricate, and it is not easy to reconstruct the author’s arguments.

For example, it is claimed that a “common mistake […] is the assumption that evolutionary trajectories form a branching tree-structure with a common root […]. However, the evolutionary topology that arises from various types of symbiosis, sexual reproduction, and a meta-configured genome (summarised below) is far more complex than any tree structure.” (lines 67-71). In support of this thesis the author quotes Lynn Margulis, according to whom “evolutionary *branching trees* need to be replaced by more complex *tangled networks*” (lines 161-162). However, according to Margulis, it is *biological* evolution that forms a tangled network, while the examples proposed by the author seem to concern cultural evolution. For example, “cross-language variation in linguistic competences” is mentioned (line 71). In the case of cultural evolution, the claim of a “tangled” evolution is almost trivial. Or is the author claiming that in some sense the tangled nature of *biological* evolution is also relevant for the problem of consciousness? The paper is not clear on this point. According to lines 163-4, it seems so, but no argument is proposed. It is suggested that the network structure of biological evolution could “imply” (line 163) the heterogeneous nature of consciousness: “forms of biological consciousness that combine different evolutionary heritages may have different functions and mechanisms, and instead of forming a branching tree of types they are likely to form a network of types without a unique ordering relation” (lines 163-165). But, in my opinion, the heterogeneous nature of consciousness and the network structure of evolution are independent facts: an untangled evolutionary tree is fully compatible with the development of multiple consciousness mechanisms, and vice versa.

The definition of the concept of Meta-Configured Genome (MCG) theory given in the first paragraph of sect. 4 (lines 138-146) is not very clear to me (examples are more effective). Considered the centrality of this notion for the argument of paper, a greater explanatory effort would be welcome. In addition, the claim that “The best known and most spectacular example of this is the huge diversity among human languages resulting not from genetic diversity so much as from diversity of cultural heritages within which genes are expressed” (lines 177-179) is dubious, since all human natural language turned out to be very similar from the point of view of linguistic structure.

The author claims that the article is “partly inspired by Turing’s 1952 paper on chemistry-based morphogenesis” (lines 20-21), in particular, if I understand correctly, by the thesis of the relevance of chemical phenomena in accounting for consciousness (see sect. 5, as far as mathematical cognition is concerned). However, it is not clear how this thesis relates to the arguments presented in the article.

Considerations concerning mathematics (which, given also the title of the article, would deserve more in-depth analysis) are certainly interesting, but it is not clear to me in which measure they are relevant to the topic of consciousness.

The list of keywords is not particularly satisfactory. For example, the pair “altricial/precocial” is mentioned in the list, but not in the text of the article; “robotic consciousness” is an item of the list, but it is not discussed in the paper.

Finally, some typos and formal details:

Line 19 – In my opinion, the spelling of “esspecially" is wrong.

Lines 116-118 - A closed bracket is missing.

At lines 256-259 the author claims “The following abbreviations are used in this manuscript: MDPI […] DOAJ […] TLA […] LD [….]”, but I could not find them in the text.

Author Response

I am very grateful for the time taken by the reviewer to read and comment on my hastily written paper produced under some duress, partly as I was waiting for a hip replacement operation -- successfully completed mid-March, I am pleased to report, though  for a while the effects held up my work on updating this paper.

I sincerely apologise for the delay and for any inconvenience this causes, especially as I am very grateful for the opportunity I have been given to submit a revised version of this paper, in which I have tried to make clearer the very broad scope and intricate variety of evolutionary and developmental mechanisms that I think need to be taken into account if we wish to produce a satisfactory science of consciousness.

The reviewer writes, in response to my reference to the work of Margulis, which is part of my inspiration: "However, according to Margulis, it is *biological* evolution that forms a tangled network, while the examples proposed by the author seem to concern cultural evolution." It is possible that I have mis-remembered or misunderstood something I have read by Margulis or about her, but in any case part of my point is indeed that there are multi-level and cross-level influences in evolution and development, that are relevant to a broad and deep science of consciousness that takes proper account of the full variety of processes of evolution and development that underpin the varieties of consciousness that need to be understood if we wish to continue the work done by Darwin, Huxley, James and others who first noticed the challenge, and the opportunity, that the theory of evolution provided as a source of new insights into the nature and varieties of states and processes labelled consciousness and closely related states and processes that are not widely recognised as part of the same general phenomenon.

In contrast, much of the recent work that claims to provide models or explanations of aspects of consciousness seems to me to focus on arbitrary fashionable subsets of the rich variety of interconnected biological phenomena that need to be explained. The new version of my paper attempts to spell out in more detail the scope and complexity of that variety (including ancient forms of mathematical consciousness discussed by Kant but nowadays mostly ignored in discussions of consciousness). While writing the revised version I tried to be more precise about the huge variety of phenomena to be explained and the corresponding variety of mechanisms. But I understand that some readers may not be convinced by the suggestion that that there are important common threads connecting all those phenomena.

The reviewer also writes "... an untangled evolutionary tree is fully compatible with the development of multiple consciousness mechanisms, and vice versa." Insofar as that is a claim about what is logically possible, I agree. My point is that in fact the evolutionary and developmental mechanisms are richly entangled and that contributes to a much richer variety of conscious phenomena than tend to be recognised in most discussions, including philosophical discussions and attempts to build explanatory models (e.g. neural network models).

The reviewer challenges what I wrote about human language, as follows:

In addition, the claim that “The best known and most spectacular example of this is the huge diversity among human languages resulting not from genetic diversity so much as from diversity of cultural heritages within which genes are expressed” (lines 177-179) is dubious, since all human natural language turned out to be very similar from the point of view of linguistic structure.

It is true that some linguists have made such claims (and some of the time Chomsky seems to be making such a claim. It can be made vacuously true by defining "linguistic structure" so that it omits all the  variation in human languages, but I think it is fair to challenge the claim by drawing attention to significant differences, including differences in the richness and variety of syntactic and semantic contents between languages, including extensions produced by mathematicians, physicists, logicians, engineers, and nowadays computer scientists whose programming languages are used and understood by humans as well as by machines. The Meta-Configured-Genome theory attempts to explain how mechanisms of delayed abstract gene expression that makes use of parameters that individuals may unwittingly have picked up earlier from the environment can produce important kinds of novelty in language, science, technology, culture, etc. that could not come from more conventional mechanisms of learning and development. I hope the expanded version of the paper helps to make this clearer, but personal experience suggests that it really requires face-to-face discussion of examples, because the ideas are so unfamiliar to most people.

The reviewer writes:

"The author claims that the article is “partly inspired by Turing’s 1952 paper on chemistry-based morphogenesis” (lines 20-21), in particular, if I understand correctly, by the thesis of the relevance of chemical phenomena in accounting for consciousness (see sect. 5, as far as mathematical cognition is concerned). However, it is not clear how this thesis relates to the arguments presented in the article."

The main point, which I hope is clearer in the revised version, is that chemical structures and processes -- as discussed by Schrodinger in "What is life?" combine discrete transitions (often produced by catalytic processes that alter bonds between particles) and continuous processes, e.g. folding, moving together or apart. (Schrodinger mainly focused on the former.) These two features seem to be capable at least in principle of supporting forms of computation that are richer than purely discrete/digital computation. The main relevance to my article is that nothing in current computational models or neural theories seems to be able to explain ancient discoveries about possible and impossible geometric structures and processes. If forms of computation that allow both discrete and continuous processes are considered they may provide novel explanations. This could be consistent with recent investigations of information processing using sub-neural chemistry. My revised paper invites Hameroff and Penrose to be more precise about how their proposed quantum processes in microtubules might have this explanatory power. So far I have an open mind about that.

The reviewer writes:

"Considerations concerning mathematics (which, given also the title of the article, would deserve more in-depth analysis) are certainly interesting, but it is not clear to me in which measure they are relevant to the topic of consciousness."

The connection is that any deep/general theory of consciousness should be able to explain the processes and mechanisms involved in ancient forms of mathematical consciousness, including the discovery processes of people like Archimedes,Euclid, Zeno and others. That was a major part of mathematical education in my youth but has recently fallen out of favour for bad reasons. Any general theory of consciousness should be able to explain what was going on when those original discoveries were made, which I think are closely relevant to consciousness of spatial (positive and negative) affordances in humans and other animals. I have tried to make the connection clearer in the revised version, but for some readers whose mathematical education has been too narrow, it may be difficult.

Finally,

I accept the reviewer's criticisms of the choice of keywords in the submitted paper, given the limited contents of the paper. I hope the longer more detailed version of the paper makes the keywords more apt.

The other (minor) points in the review have now been fixed.

Reviewer 3 Report

The aim of this paper is to study the different varieties of consciousness (biological or not) and their apparition and development during evolution.

The problem is that the term 'consciousness' is not explicitly defined and, from the given examples, would rather correspond to what is called "cognitive processes" in Neuroscience. For instance the author insists on what he calls "mathematical consciousness" (leading to higher mathematical discoveries) which would rather correspond to intuition and creativity.

That aside, the author gives a brief and interesting overview of the "Meta-Morphogenesis project", which searches to characterize how increasingly complex forms of compositionality can develop. Their evolution is not tree-directed but rather tangled (e.g. development of layers of linguistic competence); it leads to a variety of construction kits and new mechanisms using different kinds of information possibly acquired through exploratory behaviours, including semantic information and genetic information relying on a "Meta-Configured Genome".

The last section of the paper is more 'political' than 'scientific', giving a pessimistic view of the present situation (e.g. in teaching and science) in view of correcting it, with critics to AI and an appeal for new collaborators to the 'Meta-Morphogenesis project'.

Author Response

I am very grateful for the time taken by the reviewer to read and comment on my
hastily written paper produced under some duress, partly as I was waiting for a
hip replacement operation -- successfully completed mid-March, I am pleased to
report, though for a while the effects held up my work on updating this paper.

I sincerely apologise for the delay and for any inconvenience this causes,
especially as I am very grateful for the opportunity I have been given to submit
a revised version of this paper, in which I have tried to make clearer the very
broad scope and intricate variety of evolutionary and developmental mechanisms
that I think need to be taken into account if we wish to produce a satisfactory
science of consciousness.

I am aware that most of the people who write about or discuss consciousness do not attempt to address that variety, and this is reflected in the reviewer's comment:

"The problem is that the term 'consciousness' is not explicitly defined and, from the given examples, would rather correspond to what is called "cognitive processes" in Neuroscience. For instance the author insists on what he calls "mathematical consciousness" (leading to higher mathematical discoveries) which would rather correspond to intuition and creativity."

I am aware that there are many researchers who claim to be studying consciousness who think and write about only a tiny subset of the phenomena I have attempted to address. Part of my aim is to convince readers that they are focusing on an arbitrary fashionable subset of phenomena and thereby ignoring a host of related phenomena that should be part of the study of consciousness. I hope it is not too arrogant to compare that with ways in which the content of physics has been expanded from time to time (e.g. from Aristotle, to Newton, to Einstein) with loosely defined terms (e.g. force, mass) shown to have more precisely definable special cases.

The special cases of consciousness to which I have drawn attention are not arbitrary selections by me: they may have been omitted from recent fashionable writings on consciousness but were included in deeper richer writings a century earlier, including work by Darwin, Huxley, and James. In my revised version I have made this historical context much more explicit and given more detail. I have also tried to show how the current narrow fashionable investigations and models omit a wide variety of important detailed phenomena for which we need explanatory mechanisms, and which seem to be connected with a rich network of ongoing research in biology and neurosciences as well as examples of mathematical discovery that obviously involve consciousness. I hope the revised version helps, but I understand that there may remain a community of researchers who use the word "consciousness" in a narrowly restricted way and therefore find my discussion irrelevant. Perhaps they will not be convinced until the questions I've tried to address have more definite and well substantiated answers, which I admit they still lack!

Round 2

Reviewer 1 Report

It is real to re-read the same article and still be fascinated, which has been the case with my re-reading. Introductory parts added by the author may seem overly historical, even slow going, but in fact they provide the background that help follow the main argument of the article, which is very unusual but well-presented and very worth publishing. For me, this is an outstanding article.

The article requires proofreading. Including:

  1. p. 16/759 but but
  2. p. 18/815-816 (proofreading required

Groundbreaking ideas seem especially important, and well presented, in those places:

  1. 11/508 “What all organisms (including sub-microscopic organisms invading other organisms) have in common is acquisition and use of semantic information” Controversial but well-presented and scientifically sound thesis.
  2. 17/763-770 Very important ideas – potentially groundbreaking

While the author should be ale to participate in proofreading, including on p. 18 -- according to this reviewer the article does not require further corrections (beyond careful proofreading)

Reviewer 3 Report

The author has made important revisions to his first version which answer to the different critics I had mentioned. In particular he now well explains what he means by "consciousness" and the paper is interesting to read.

I just regret his pessimistic view of current mathematics opposed to ancient mathematics. There are still "discoveries about properties of collections of discrete objects and properties of counting operations and one-to-one correspondences", except that the objects are now diagrams and construction and operations about them instead of lines, circles...

This manuscript is a resubmission of an earlier submission. The following is a list of the peer review reports and author responses from that submission.